# Long-Term Ecosystem Nutrient Pool Status for Aspen Forest Harvest Simulations on Glacial Till and Sandy Outwash Soils †

**Robert P. Richard** [1,*] , **Evan S. Kane** [1,2] , **Dustin R. Bronson** [3] **and Randall K. Kolka** [4]

1 College of Forest Resources and Environmental Science, Michigan Technological University, U.J. Noblet Building, 1400 Townsend Drive, Houghton, MI 49931, USA; eskane@mtu.edu
2 U.S. Forest Service—Northern Research Station, 410 MacInnes Drive, Houghton, MI 49931, USA
3 U.S. Forest Service—Northern Research Station, 5985 Highway K, Rhinelander, WI 54501, USA; dustin.bronson@usda.gov
4 U.S. Forest Service—Northern Research Station, 1831 Hwy 169 East, Grand Rapids, MN 55744, USA; randall.k.kolka@usda.gov
* Correspondence: rprichar@mtu.edu
† This work was part of the doctoral dissertation of the first author Robert Richard. College of Forest Resources and Environmental Science in Michigan Technological University, Houghton, MI, USA.

**Abstract:** Sandy outwash and glacial till soils compose large amounts of public forestland due to historically poor agricultural yields. The outwash soils have low fertility, poor nutrient retention and are restricted from whole-tree harvesting (WTH) in some states, whereas the glacial till has medium nutrient retention and fertility, and is unrestricted from WTH. To assess the long-term sustainability of harvesting, a nutrient budget was constructed from field measurements, the National Cooperative Soil Survey (NCSS) database, and literature values for stem-only harvesting (SOH) and WTH at a 45-year rotation length and 11 rotations were simulated. The budgets showed that SOH and WTH recovery years, or the time necessary for the inputs to match outputs through leaching and one harvest, exceeded common rotation lengths for both soil types under all weathering scenarios, and the average WTH reduced the total available rotations by one harvest. The large variation in soil nutrient pools and harvest removals complicated the ability to identify the difference between SOH and WTH early in the model, but differences became apparent with sequential harvests. The recovery years were 2–20 times the 45-year rotation length under all weathering rates. Taken together, models in this study bridge the gap between short- and long-term studies and bring into question the sustainability of WTH and SOH practices on nutrient-poor soils.

**Keywords:** whole-tree harvesting; nutrient budget; harvest intensity; ecological modeling; soil sustainability

## 1. Introduction

Nutrient-poor and medium fertility soils, such as glacial outwash (Typic Udipsamments, Entic Haplorthods) and tills (Alfic Haplorthods), are abundant throughout the Great Lakes region, USA. The soils compose large percentages of managed public forestlands after being abandoned as the "lands nobody wanted" [1] and are commonly managed in a regime of even-aged aspen clearcut harvesting. Bigtooth aspen (*Populus grandidentata* Michx.) and quaking aspen (*Populus tremuloides* Michx.) are the most widely distributed trees in North America [2] and contain large amounts of Ca and Mg in woody tissues [3]. However, there is still much uncertainty as to the frequency of harvests that can be conducted in lower to medium fertility soils and the effects of harvest on long-term available nutrient stocks [4–11].

The shift towards including woody biomaterials in the economy has increased the demand for woody biomass yield in forest harvesting practices [12]. Intensive forest harvesting is a concern for nutrient sustainability in low fertility or nutrient-poor soils and

therefore is restricted in some US state management plans [8]. Whole-tree harvesting (WTH) differs from conventional stem-only (SOH) forest harvesting due to the inclusion of smaller woody tissues and branches containing higher concentrations of macro-and micronutrients than bolewood used in dimensional lumber production [4,13,14] and therefore may prove detrimental to long-term forest productivity.

While it is assumed that repeated removals of nutrient-dense tissues could have detrimental effects on long-term soil productivity in coarse-textured soils [7], few forest nutrient budgets exist to ascertain the long-term effects of repeated timber harvest [15]. In nutrient budget studies in aspen-dominated forests with relatively nutrient-rich soils, pools of nitrogen (N), phosphorus (P), and potassium (K) removed in harvest were estimated to be replaced by inputs, whereas magnesium (Mg) and calcium (Ca) were a concern for depletion [4,5]. Calcium was depleted within 9–30-year rotations [4], and Ca and Mg would not replenish the pool size on a 60-year rotation length [5]. Among different forest types, the long-term productivity declines have called into question the iterability of the WTH practice without supplemental inputs as more base cations were removed than were supplied by weathering and deposition [6,16].

Observations into the short-term ($\leq 1$ rotation) soil pool response on aspen-dominated ecosystems have shown mixed and highly variable results between SOH and WTH [9–11,17,18]. In operational forestry, the residual pool of soil Ca following WTH was approximately half that of SOH on an outwash soil (Entic Haplorthods) [9], although pools of Ca, Mg, and K were not significantly different [10]. A more nutrient-rich moraine soil (Inceptic Hapludalfs) showed larger soil K pools on sites after WTH than SOH, and Ca and Mg were not significantly different between harvest methods [9,10,18]. Adding to this complexity, the season of operability for both WTH and SOH can affect the magnitude of the pool of nutrients removed, with more nutrients being stored in aboveground tissues during the growing season [19]. However, complicating factors such as nutrient translocation and changes in inputs need to be considered to detect the effects of harvest removal through ecosystem observations [10,11].

The outputs of a managed forest ecosystem are typically harvested removals and leaching, and the inputs are weathering and atmospheric deposition, but specific inputs also vary by site. For example, groundwater inputs can replace the nutrients lost to intensive harvests [10,20]. In the absence of the additional input term or supplemental fertilizers, intensive forest harvest removals can greatly exceed the inputs from weathering and deposition [6]. These studies demonstrate the importance of understanding site-specific differences in input terms to the ecosystem, from groundwater, soil texture and corresponding soil weathering rates, and deposition.

Given the long-term productivity concerns and changes in site-specific results, edaphic conditions are increasingly emphasized in forest management recommendations. Both Connecticut and Wisconsin have incorporated soil map units in WTH management policy [8,21]. In Connecticut, the variables of interest include drainage class, water capacity, depth to the water table, depth to restrictive layer, cation exchange capacity (CEC), soil organic matter (SOM), erodibility, and slope [21]. Wisconsin also relies on soil map unit interpretations and restricts WTH on soils by clay content, CEC, drainage class, absence of lamellae, and carbonates [8]. Michigan and Minnesota address long-term productivity by the recommendation to leave residual biomass [22,23]. Although state agencies have addressed potentially negative WTH effects on long-term soil productivity through biomass retention cutoff percentages, there is little consensus as to the aggregate effects of multiple harvests.

The mixed results and differences between the short and long-term studies leave a knowledge gap regarding forest management policy for a fairly straightforward question: Can woody biomass be harvested sustainably in nutrient-poor soils? The goal of this study was to model the known difference between SOH and WTHs [19] in combination with the inputs and outputs through time to assess the long-term effects on aspen-dominated ecosystems with sandy outwash and glacial till soils. The secondary objective was to

examine seasonal changes in the effects of WTH to determine if there are seasons of operability wherein nutrient removals can be minimized due to observed differences in residuals from breakage [19]. To meet these objectives, seasonal input–output budget models of forest tissue and soil elemental pools were constructed. The harvest models considered inputs and outputs on a 45-year rotation length and compared SOH to winter, spring, summer, and fall WTHs. Each harvest was repeated over time to determine the long-term effects of the harvest. Recovery years, defined as how long it would take for inputs to replenish values lost to one harvest and leaching, were calculated for comparison with typical rotation lengths.

## 2. Materials and Methods

### 2.1. Overview

This study examined nutrient inputs and outputs in aspen forests occurring on harvest-restricted soil complexes in Wisconsin, USA. We leveraged soil data collected from 14 new soil pedons (Figure 1), national soil databases, and literature values (see appendix for details). A nutrient budget was constructed for the soils considering inputs as weathering and deposition and outputs of soil leaching and harvest removals in comparison to available nutrient pools similar to previous research [5,6]. Soil weathering has previously been measured on the sandy outwash and till soils considered in this study, and the depletion method rates were used for the annual weathering inputs [24]. The total deposition was derived from deposition maps ([25]; Figure A1). Soil and tree tissue information was collected across the northern portion of the state of Wisconsin (Figure 1). The nutrient budget considered SOH and WTH during the winter, spring, summer, or fall, under three regional deposition budgets and three weathering scenarios.

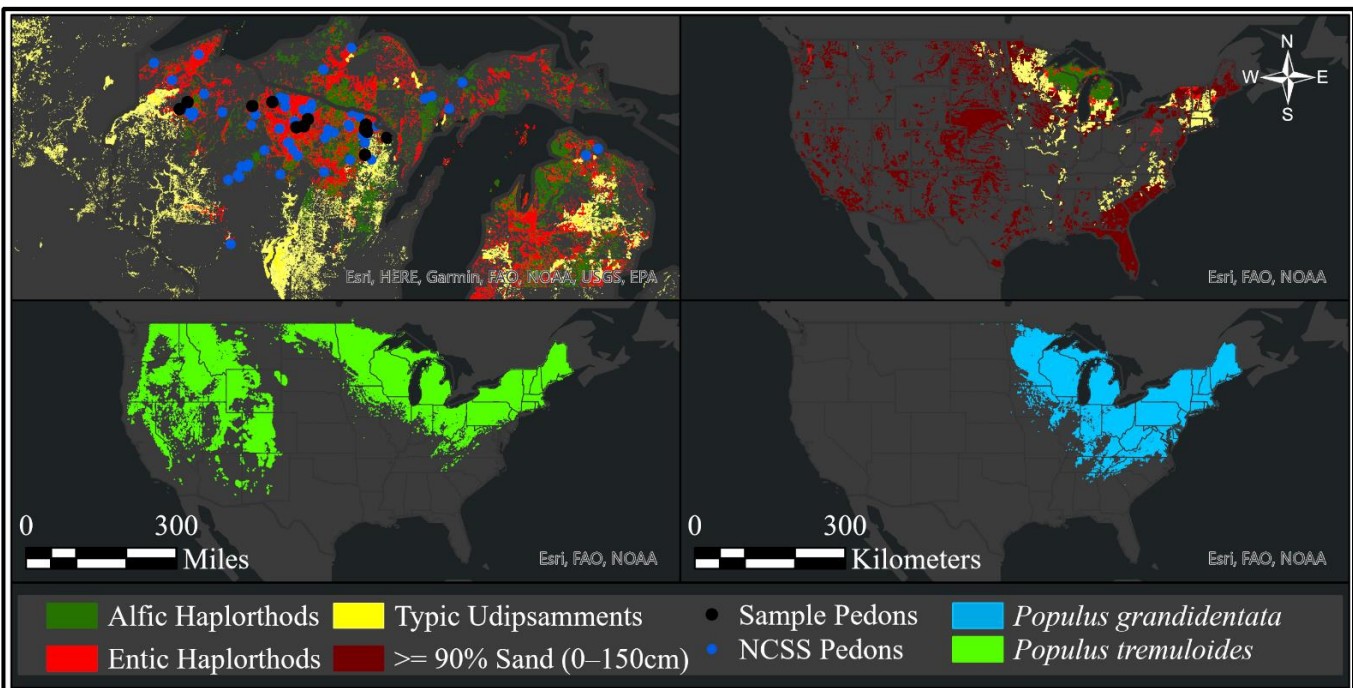

**Figure 1.** The major nutrient-poor outwash (Typic Udipsamments-Entic Haplorthods), glacial till (Alfic Haplorthods), and low silt + clay soils from the gridded Soil Survey Geographic (gSSURGO) database. The sample pedons were collected with NRCS soil scientists to increase available pedons in outwash soils, and NCSS pedons were retrieved from the database for the outwash and glacial till soils. The aspen species within this study have large ranges across North America and are shown within the United States (derived from [26]).

*2.2. Field Sampling*

Site selection was limited to aspen forest stands from the Wisconsin Forest Inventory and Reporting System (WisFIRS) within the 40–55-year harvest window. Sites were located on Entic Haplorthods or Typic Udipsamments soil map units from the gridded Soil Survey Geographic (gSSURGO) that are currently restricted from WTH [8]. Soil pedons were sampled cooperatively with Natural Resource Conservation Service (NRCS) soil scientists during the summers of 2016 and 2018 (Figure 1), following standard procedures, with bulk densities taken in triplicate [27]. The 2018 pedons were also sampled for leachates from 150–160 cm during the summer (12 July 2018) and resampled by bucket auger from 150–160 cm during winter (pre-frost layer, 8 January 2019) and spring (snow-on conditions, 12 March 2019) to capture the seasonal variation of water-soluble nutrients below the rooting depth. The below-rooting depth was chosen as previous research has defined the rooting depth of aspen on outwash soils at 150 cm [24]. The nearest dominant or codominant aspen tree to the soil pit was sampled for nutrient content from 18–25 June 2018 and on 11 September 2018. Each tree was felled and sampled for a basal disc, a disc at diameter at breast height (DBH), bolewood, barkwood, live branch, dead branch, twig, and leaf.

*2.3. Laboratory Analysis*

The soil samples were transported to Michigan Technological University, Houghton, MI. Leachate concentrations were obtained by shaking 20 g of field moist soil with 100 mL of deionized water for one hour and filtered through a Whatman 42 filter and a 0.45 μm filter [28,29]. The P, K, Ca, and Mg concentrations were determined with a Perkin Elmer Optima 7000 DV Inductively Coupled Plasma Optical Emission Spectrometer ICP-OES, and the N was obtained by Shimadzu DOC/TN analyzer. Soil pool values for the outwash soil sampled with the NRCS were obtained from the results of the USDA-NRCS NSSC National Soil Survey Laboratory. Soil pool values were also obtained from the National Cooperative Soil Survey (NCSS) database. Total N was determined by method 4H2a (dry oxidation), extractable P by method 4G4, and extractable Ca, Mg, and K were determined by method 4B1a1a (NH4OAc and 2 M KCl rinse) [30]. Total P, Ca, Mg, and K were obtained by ashing in a muffle furnace to remove carbon from the sample for 8 h at 500 °C and then acid digested by EPA 3052 method and processed by a Perkin Elmer Optima 7000 DV Inductively Coupled Plasma Optical Emission Spectrometer (ICP-OES). Missing values were imputed through kNN imputation in the Caret package [31] in the R statistical platform at the horizon level, and the weighted average by horizon depth was calculated for a depth of 150 cm.

The tree tissue samples were transported to the Wisconsin Department of Natural Resources, Forestry Headquarters, Rhinelander, WI, and dried at 55 °C to a constant mass. Tissues for N analysis were ground in a ball mill for 5 min, equipment cleaned with ethanol between samples, and 5 g were analyzed on a Costech 4010 Elemental Analyzer. For nutrients P, K, Ca, and Mg, tissues were ground in a Wiley mill, acid digested by EPA 3052 method, and processed by a Perkin Elmer Optima 7000 DV Inductively Coupled Plasma Optical Emission Spectrometer (ICP-OES). The DBH disc was used for both bolewood and barkwood samples. The barkwood was sampled from the outer to the inner bark, and bolewood was sampled from the inner bark to the center ring. The branches were sampled with a cross-section taken from the center of the branch, with care taken to ensure the cross-section remained intact. The twigs and leaves were sampled in their entirety. Roots were grouped into coarse (≥5 mm) [32] and fine roots (<5 mm) and pooled by soil genetic horizon type (A-AE, E-EA, Bhs, Bs1, Bs2, Bs3, BC, C1, C2) to obtain enough volume for nutrient analysis.

*2.4. Harvest Intensity Simulations*

2.4.1. Ecosystem Inputs

The two main nutrient inputs for forest ecosystems are soil weathering and atmospheric deposition. Soil weathering has been estimated by several methods, and the depletion method is recommended for use as there is more confidence in the assumptions and it has the least uncertainty [24]. The depletion method measured the texture classes of the soils, conducted elemental analysis by X-ray diffraction, converted to elemental percent, and used multiple regression to calculate depletion [24]. The minimum, average, and maximum weathering rates for Ca, Mg, and K from the depletion method located on outwash soils [24,33] were used to create linear regression equations between elemental concentrations and sand percentages, except for the minimum K equation, which was non-linear and required exponential regression. The weathering rates of outwash soils have been related to the aeolian mass; hence the equations were necessary to adjust the weathering rates from the depletion method for Ca, Mg, and K to the average soil texture observed within the soil pits (Table A1). The texture-adjusted minimum, average, and maximum weathering rates were used as inputs into the nutrient budget for the observed soils sampled with the NRCS. The average weathering values from the outwash soils were used for the NCSS outwash soils and the Warba series for the NCSS Alfic Haplorthods [24]. The weathering rates from both the texture-adjusted and non-texture-adjusted outwash soils were used as model parameters to calculate recovery years after a harvest. Since the textural variation has been observed previously on the outwash soils [24], we treated the weathering rate as a parameter and considered the minimum, average, and maximum weathering rates per element.

Atmospheric deposition was comprised of wet and dry deposition. Wet atmospheric deposition has been measured since 1978, and total atmospheric deposition has been estimated from 2000 to the present [25]. The deposition amounts vary spatially but generally show a pattern of lower deposition in the northeastern part of the state of Wisconsin and higher in the south and southwestern portion within an example 45-year rotation length (Figure A1). Given the differences in deposition amounts, the nutrient model was calculated for three regions. The regions were northeast, northcentral, and northwest, corresponding to the dominant advancement and retreat patterns from the last glaciation [34]. The deposition values for N, K, Mg, and Ca were extracted from the total deposition raster files [25] by the sampling points for years 2000–2017, and an average value per region was calculated per year. The deposition time series were transformed, detrended, and forecasted annually to the end of the rotation length (45 years) using the R libraries *t*-series [35], TSA: Time Series Analysis [36], and forecast [37,38]. The augmented Dickey–Fuller test was performed to ensure no unit root existed after transformation and detrending, which would result in a skewed forecast. The total deposition values were best detrended by differencing and log transformation (Figure A2; Table A2). Typically, the series passed or nearly passed the adjusted Dickey–Fuller test ($\alpha = 0.05$), except for the K in the northeast and northcentral areas (Table A2). However, across all nutrients, there was a spike in the 2017 values, which appeared to cause the K time series to fail since the removal of the 2017 value results in generally white-noise or detrended processes (Figure A2). Since the transformed K series appeared to be white-noise and other transformations also did not pass the Dickey–Fuller tests, the differenced and log-transformed values were used to forecast the K time series. The annual observed values from 2000–2017 combined with the forecasted values for one rotation were used for repeated rotations.

2.4.2. Ecosystem Outputs

The two main nutrient losses from the forest ecosystem are leaching and harvest removals. Leaching rapidly increases after harvest and slows to the undisturbed state after approximately 5–10 years [5,39]. To approximate this post-harvest effect as a time series, the undisturbed leaching rate must first be provided by the field observed values. The undisturbed leaching rate was established by multiplying the observed seasonal nutrient

concentrations by the outflow of a local water balance. The water balance was calculated monthly and was comprised of inflow as precipitation minus output from evapotranspiration. The change in water storage, as determined by the water holding capacity of the soil pedons, was used to complete the water balance [28,40]. Evapotranspiration was calculated by monthly average temperature and hours of daylight, as determined by latitude [41]. The undisturbed leaching values could then be modified by leaching ratios at 1, 2, and 5 years post-harvest [5], with the weighted average being applied between the available times. The control (CON) assumed no biomass removal and used the undisturbed leaching rate.

Decisions about harvest removals must consider differences between the nutrients removed from SOH and WTH, which determine the volume of removals by tissue. The starting forest volume available for harvest removals was calculated by scaling the sampled tissues per tree on a per hectare basis. The sampled tree tissue volumes were normalized to 45 years and scaled allometrically by tissue type; bolewood, barkwood, live branch, leaf, twig [42], dead branch [43], coarse roots [32] and stump, and large and medium roots [44]. The volume per tree was scaled to trees per hectare by the average basal area and site index of aspen stands located within the rotation length window (40–55 years) and on restricted soils in the stocking guide [45]. The available forest volume for harvest was then modified by harvest type, with SOH consisting of the merchantable volume of bolewood and barkwood [44] and WTH removing approximately 30% more fine woody debris (FWD) and 60% more coarse woody debris (CWD) [19]. The foliar and twig volumes were considered as FWD, and the live and dead branches were considered as CWD. The winter WTH was modified as it was found to leave 33% more FWD on-site than the remaining WTHs due to increased breakage [19]. Seasonal changes in tissue nutrient concentrations were applied to these volume estimates in determining how the harvest type affected changes in nutrient removal in different harvest times.

The observed tissue concentrations were multiplied by the volume of each tree component to provide the number of macronutrients removed by harvest type. The FWD and CWD concentrations were also modified by the ratios between the seasonal differences. Seasonal differences were determined by the average of the season for foliar (leaf + bud) and twig concentrations [13,46–49]. The seasonal fluctuations and the seasonal averages across studies are presented for the foliar and twig nutrient concentrations (Appendix A Figures A2 and A3).

### 2.4.3. Nutrient Capital and Modeling Assumptions

The harvest intensity simulations considered losses to the system from leaching (OL) and harvest removals (OH) and inputs to the system from total deposition (ID) and weathering (IW) on an annual basis.

$$ID + IW - OL - OH = \text{Nutrient Balance}$$

Six harvest types were considered: CON, SOH, and seasonal WTHs (Winter, Spring, Summer, Fall). The CON did not remove biomass, and the SOH scenario did not include seasonal changes because the branches were left on site. The WTH harvest type required the twig and leaf-bud values to be adjusted to the seasonal averages by the seasonal ratios and sampling time [13,46–49] and differences in volume removed [19]. The initial soil value is the starting nutrient capital from which the inputs and outputs can be compared annually. The soil pool was defined as the top 150 cm, and soluble nutrients below this depth are considered below the primary rooting depth [24]. The unharvested volume composed of root and stump values was then combined with the soil values to constitute the nutrient capital post-harvest. The nutrient budget was calculated on a rotation length of 45 years which is a common rotation length of aspen in the region [50]. Each rotation included one harvest year, and the harvests were repeated until the capital pool reached depletion or up to 11 rotations (495 years). Confidence intervals were calculated for the harvest removals and soil capital ($\alpha = 0.05$). The maximum soil and minimum harvest bounds were used to create an ecosystem upper bound, and the minimum soil and maximum

harvest bounds were used to create a lower bound for the ecosystem. Although a forest stand will likely experience different types and seasons of harvest, the goal of this study was to evaluate sustainable harvest scenarios from a soil element "recovery" perspective. Recovery years were calculated by the time it took for the annual outputs and inputs of an element to replace the values lost to one harvest. The average atmospheric deposition per year across the three modeled locations was used for the deposition input, and the undisturbed leaching value was used for the output 45 years and later. The recovery years were calculated for comparison with typical rotation lengths to observe the harvesting rate under several weathering scenarios.

## 3. Results

### 3.1. Ecosystem Nutrient Distribution

The soil contained the largest nutrient stocks in these forest ecosystems (Table 1). The sampled outwash pedons held a larger amount of K and lower amounts of Ca and Mg than the NCSS pedons (Table 1). The NCSS Alfic Haplorthods pedons, by comparison to the outwash soils, held much larger P, Ca, and Mg, similar N, and smaller K values (Table 1). The vegetation contained the second largest ecosystem nutrient values (Table 1). The vegetation on outwash soils contained a large proportion of ecosystem K, Ca, and Mg, whereas only the vegetation K values on the Alfic Haplorthods contained a large percentage of this cation (Table 1). The N and P within the vegetation pools were small in comparison to outwash and Alfic Haplorthods soils (Table 1). The largest pools of nutrients within the vegetation tended to be in bolewood, barkwood, leaf, live branches, and stump combined with large and medium roots across all reported nutrients. In comparison, the coarse roots, dead branches, and twigs tended to harbor much less of the overall nutrients within the ecosystem (Table 1). Foliar tissues showed a spike in N, P, and K during the growing season, whereas Ca and Mg showed a peak during the dormant season (Figure A3). Twig nutrient concentrations generally decreased during the growing season and had larger concentrations during the dormant season (Figure A4). Winter P values were not available, and the twig concentrations are more limited than the foliar, particularly in the winter, where only one study presented values.

**Table 1.** Distribution of nutrient pools in an aspen-dominated forest located on nutrient-poor soil types and confidence intervals ($\alpha = 0.05$). Average vegetation nutrient pools are adjusted to 45 years old in the summer season.

| | Nutrient (kg ha$^{-1}$) | | | | |
|---|---|---|---|---|---|
| | N | P | K | Ca | Mg |
| Leaf | $41 \pm 8$ | $5 \pm 1$ | $23 \pm 4$ | $27 \pm 7$ | $5 \pm 1$ |
| Twig | $0.92 \pm 0.23$ | $0.22 \pm 0.05$ | $0.99 \pm 0.14$ | $2.14 \pm 0.34$ | $0.22 \pm 0.05$ |
| D.Branch | $1.2 \pm 0.71$ | $0.15 \pm 0.08$ | $0.65 \pm 0.45$ | $6.18 \pm 4.30$ | $0.39 \pm 0.23$ |
| L. Branch | $34 \pm 10$ | $7 \pm 2$ | $38 \pm 11$ | $136 \pm 41$ | $11 \pm 3$ |
| Barkwood | $91 \pm 28$ | $13 \pm 5$ | $94 \pm 40$ | $467 \pm 170$ | $24 \pm 8$ |
| Bolewood | $247 \pm 209$ | $23 \pm 6$ | $109 \pm 33$ | $312 \pm 158$ | $39 \pm 9$ |
| Stump + Roots | $71 \pm 48$ | $8 \pm 2$ | $40 \pm 15$ | $132 \pm 69$ | $13 \pm 3$ |
| Coarse Roots | $0.06 \pm 0.01$ | $0.02 \pm 0$ | $0.02 \pm 0$ | $0.2 \pm 0.02$ | $0.02 \pm 0.00$ |
| Vegetation Total | $487 \pm 217$ | $57 \pm 8$ | $306 \pm 55$ | $1083 \pm 246$ | $93 \pm 13$ |
| Sampled Soil Pedons (0–150 cm) | $8734 \pm 506$ | $1953 \pm 175$ | $989 \pm 123$ | $3217 \pm 339$ | $336 \pm 39$ |
| Typic Udipsamments and Entic Haplorthods NCSS Soil Pedons (0–150 cm) | $10781 \pm 996$ | $2085 \pm 260$ | $722 \pm 65$ | $3763 \pm 399$ | $527 \pm 67$ |
| Alfic Haplorthods NCSS Soil Pedons (0–150 cm) | $10743 \pm 433$ | $4398 \pm 109$ | $698 \pm 90$ | $12740 \pm 1131$ | $2988 \pm 519$ |

### 3.2. Modeled Harvest Removals

Not surprisingly, WTH removed a larger portion of all nutrients than the SOH due to more volume being removed (Table 2, Figure 2a). The winter season removed the lowest amounts of macronutrients for WTH (Table 2). The remaining seasons had different impacts by element, with the spring harvest removing larger amounts of N and K, the summer harvest removing larger amounts of K and Ca, and the fall harvest removing the largest amount of Ca (Table 2). The P and Mg pools did not differ seasonally among WTHs, but the WTH removals are larger than the SOH (Table 2). The harvest removals were one or two orders of magnitude larger than the leaching outputs depending on the nutrient, showing the degree to which the harvest removals drive the scenario outputs (Table 2). The inputs did not vary greatly since the deposition values for K and Mg are similar to the average weathering rates across regions, but the Ca deposition values were larger than the weathering rates (Table 2). The maximum weathering rates were larger than deposition inputs across all elements (Table 2).

**Table 2.** Harvest simulations macronutrient inputs and outputs of a single rotation, as well as weathering rates and deposition by region.

| | Nutrient (kg ha$^{-1}$ 45 year$^{-1}$) | | | | | |
|---|---|---|---|---|---|---|
| | | **N** | **P** | **K** | **Ca** | **Mg** |
| **Outputs** | | | | | | |
| *Harvest Removals* | SOH | 339 | 36 | 204 | 779 | 63 |
| | WTH—Winter | 396 | 48 | 257 | 977 | 81 |
| | WTH—Spring | 461 | 51 | 280 | 985 | 83 |
| | WTH—Summer | 433 | 52 | 282 | 998 | 84 |
| | WTH—Fall | 421 | 49 | 267 | 1034 | 87 |
| *Leaching* | | 4 | 0.4 | 24 | 41 | 13 |
| **Inputs** | | | | | | |
| *Deposition* | Northeast | 241 | | 9 | 88 | 12 |
| | Northcentral | 226 | | 9 | 85 | 11 |
| | Northwest | 268 | | 10 | 100 | 12 |
| *Weathering* | | | | | | |
| Entic Haplorthods/Typic | Minimum | | | 4 | 0 | 1 |
| Udisamments | Average | | | 5 | 36 | 9 |
| | Maximum | | | 8 | 70 | 17 |
| Medium to Nutrient-Rich | Warba | | | 9 | 280 | 38 |
| **Net** | | | | | | |
| | SOH | −98 | −36 | | | |
| | WTH (avg.) | −187 | −50 | | | |
| Entic Haplorthods/Typic Udisamments | | | | | | |
| *Minimum Weathering* | SOH | | | −214 | −729 | −63 |
| | WTH (avg.) | | | −282 | −948 | −84 |
| *Average Weathering* | SOH | | | −213 | −693 | −55 |
| | WTH (avg.) | | | −281 | −912 | −76 |
| *Maximum Weathering* | SOH | | | −210 | −659 | −47 |
| | WTH (avg.) | | | −278 | −878 | −68 |
| Medium to Nutrient-rich | | | | | | |
| *Warba Weathering* | SOH | | | −209 | −449 | −26 |
| | WTH (avg.) | | | −277 | −668 | −47 |

The bold and italics help to differentiate the catagorical differences.

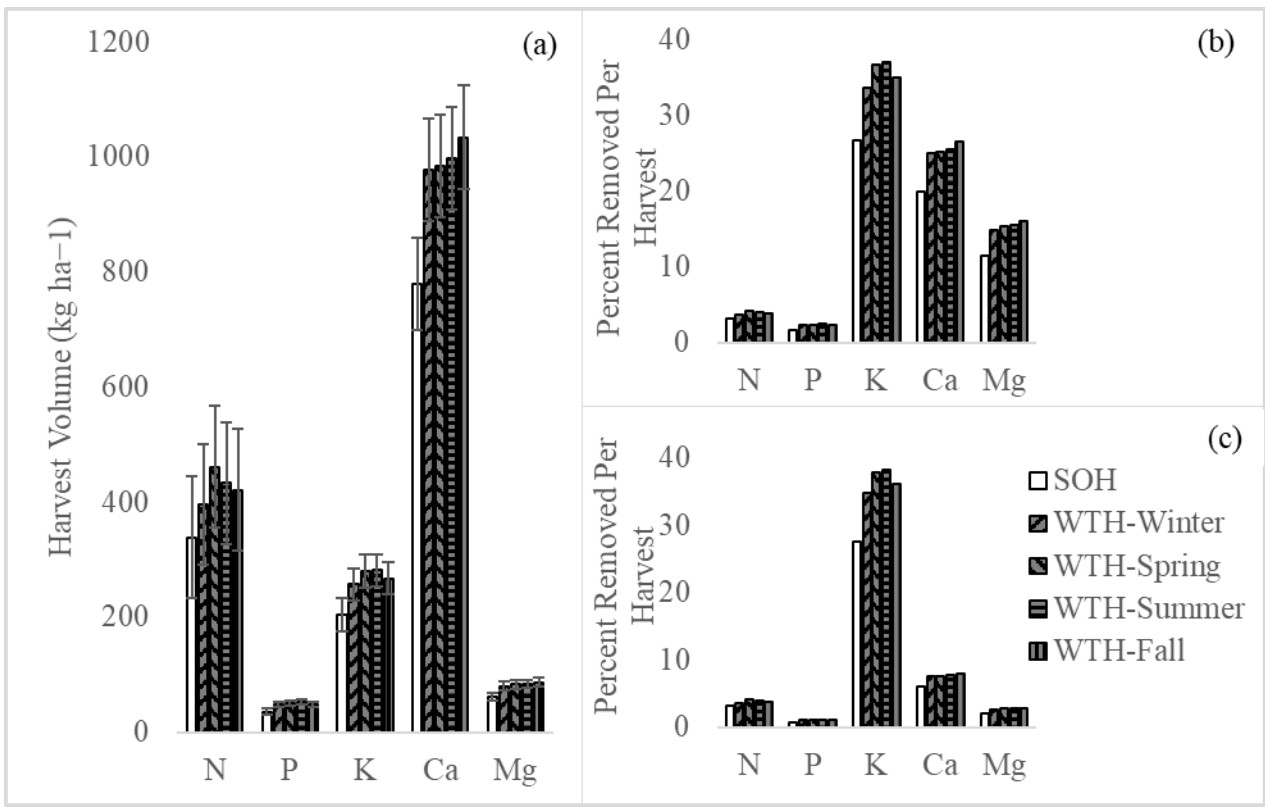

**Figure 2.** Nutrient pools removed by harvest type by (**a**) total volume, and compared to the remaining tree and root tissue values, and (**b**) outwash soils (Typic Udipsamments and Entic Haplorthods) (**c**) Alfic Haplorthods soil pools. The large proportion of K, Ca, and Mg removed per harvest from the outwash soils highlighted the nutrient-poor status of the soils. The Alfic Haplorthods showed a much smaller percent of Ca and Mg per harvest, but K remained large. N and P constituted less than ten percent of the soil pools per harvest.

When the harvest mass was compared to the remaining ecosystem nutrients for the outwash soils, the percentage of removal was approximately 5% or less of N and P, and between 20–35% of K, Ca, and Mg (Figure 2b). The harvest mass for the Alfic Haplorthods was approximately 5% or less for N, P, and Mg, and 10% or less for Ca, but remained within 30–40% for ecosystem K. In comparison to SOH, WTH removed an additional 5–10% of the ecosystem K, Ca, and Mg, and 1–2% of the N and P in the outwash ecosystems. On the Alfic Haplorthods ecosystems, the WTH removed an additional 1–2% N, P, Ca, and Mg and 7–10% K compared to SOH.

### 3.3. Harvest Time Series—Sampled Outwash Pedons

Although the deposition values varied spatially, the harvest removals drove the trend in the time series models. The time series from the northeastern model for the sampled NRCS outwash soils under the texture-adjusted average weathering scenario showed minimal impacts to N and P but rapid depletion of K, Ca, and Mg (Figure 3). The sampled outwash pedons N and P pools showed negligible harvest impacts (Figure 3a,b), and the exchangeable K and Ca was depleted on average in 4 WTHs and 5 SOHs (Figure 3c,d) and Mg depleted in 5 WTHs and 6 SOHs (Figure 3e). In comparison to the harvests, the unharvested scenario (CON) showed gradual losses in P and K (Figure 3b,c) and accumulation in N, Ca, and Mg (Figure 3a,d,e). The total soil pool size was modeled to decrease through time as the weathering occurs (Figure 3a–e).

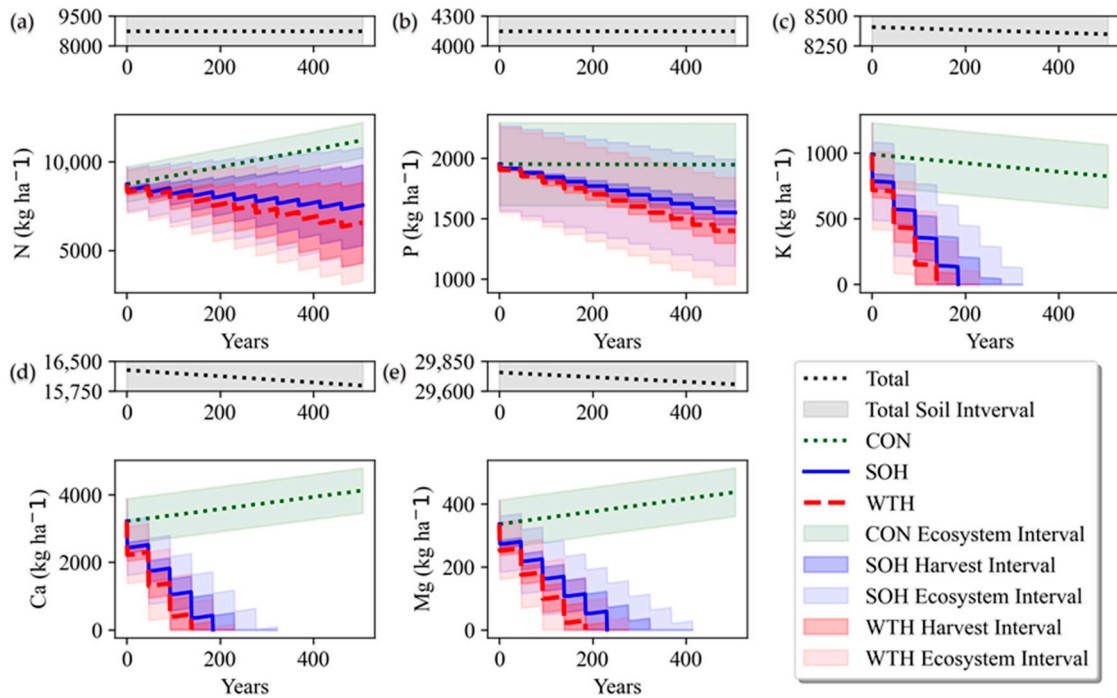

**Figure 3.** Simulated soil nutrients by scenario for the sampled NRCS pedons with texture-adjusted weathering rates. The N and P showed lower harvest impacts (**a**) N are totals (exchangeable unavailable) and (**b**) P total is unchanged (weathering unavailable). The depletion of K and Ca (**c**,**d**) is reached by 5 SOHs and 4 WTHs, and the Mg (**e**) is depleted in 6 SOHs and 5 WTHs. The control (CON) shows (**a**,**d**,**e**) accumulation in N, Ca, and Mg (**b**,**c**) gradual losses in P and K. Confidence intervals show large variability ($\alpha = 0.05$).

### 3.4. Harvest Time Series—NCSS Pedons

The seasonal time series from the northeastern model for the NCSS outwash soils under the average weathering scenario (Figure 4) and the NCSS Alfic Haplorthods (Figure 5) showed different impacts by site. The NCSS outwash pedons showed minimal harvest impacts on N and P, although differences between SOH and WTH appeared after 5 rotations. The average remaining element pools were depleted between 3–10 rotations, varying by harvest treatment: K (rotations: 4 SOH; 3 WTH), Ca (rotations: 6 SOH; 5 WTH), Mg (rotations: 10 WTH) (Figure 4). The NCSS Alfic Haplorthods pedons showed only the average K was depleted (rotations: 4 SOH; 3 WTH), and N, P, Ca, and Mg manifested differences between SOH and WTH halfway through the model. Seasonal differences appeared but rarely changed the number of total harvests and only near the end of the model.

In addition to the time series, the recovery years also showed the harvest outputs were greater than inputs at the current rotation length under all scenarios and soil types (Table 3). The K, Ca, and Mg recovery years exceed the average rotation length by a factor of two to four under the maximum weathering values on the non-texture-adjusted soils [24]. When the weathering rates were adjusted to textures observed in the field, the recovery years increased greatly and further demonstrated the non-sustainable removal rates for the outwash soils (Table 3). The Alfic Haplorthods showed recovery years approximately double the average rotation length for SOH of Ca, SOH, and WTH of Mg, and triple of WTH for Ca (Table 3). SOH and WTH removals of K were not replaced through weathering and deposition under all scenarios except for the maximum outwash weathering rate, and even under that scenario vastly exceeding an average rotation length.

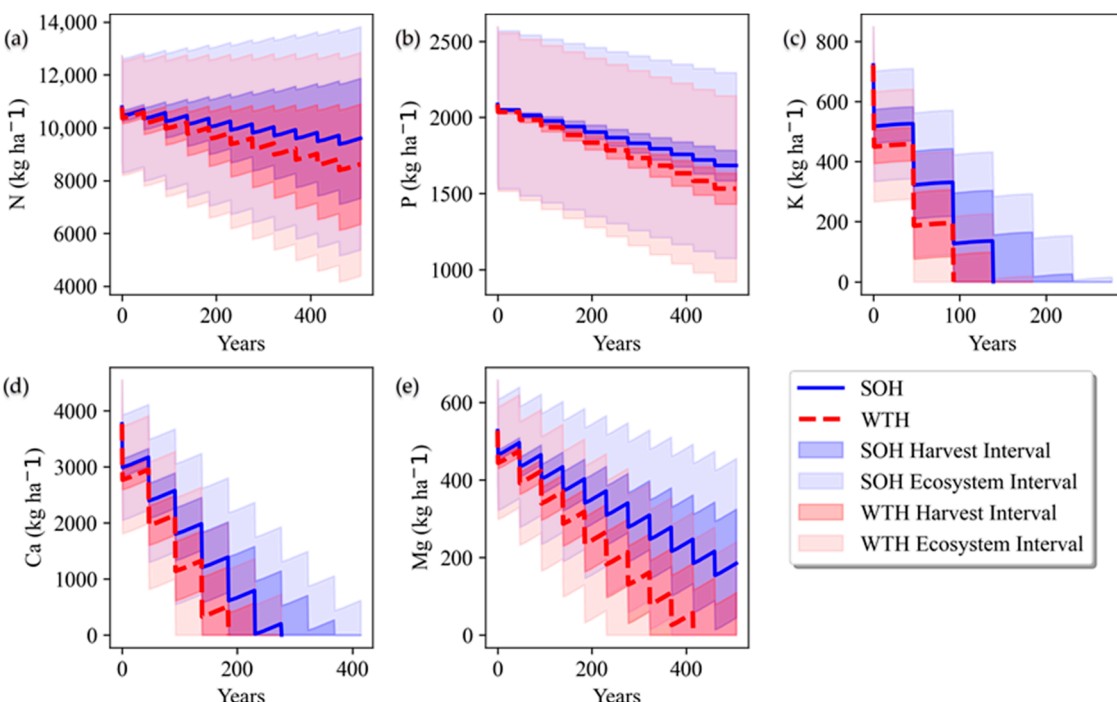

**Figure 4.** NCSS Entic Haplorthods and Typic Udipsamments pedons in northeastern macronutrient time series by harvest simulation under average weathering scenario. (**a**–**e**) The average WTH decreased total rotations across nutrients compared to SOH, although confidence intervals show large variability in the systems ($\alpha = 0.05$). (**a**,**b**) The N and P pools showed small harvest impacts, (**c**) K depleted in 3 WTHs and 4 SOHs, (**d**) Ca depleted in 5 WTHs and 6 SOHs, and (**e**) Mg depleted in 10 WTHs.

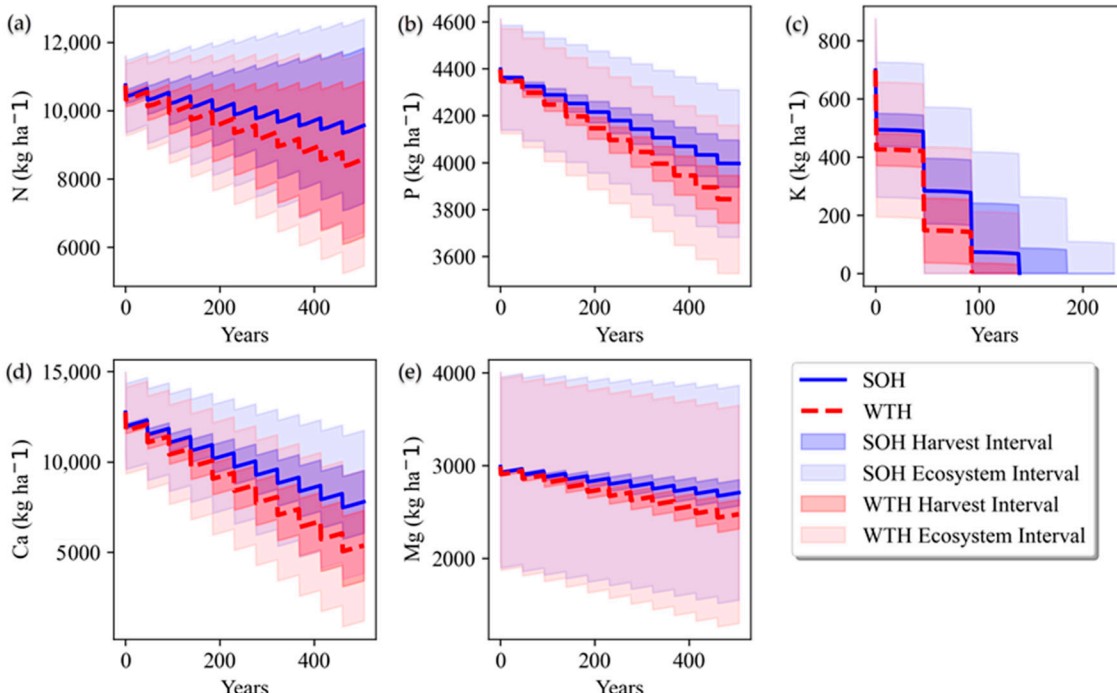

**Figure 5.** NCSS Alfic Haplorthods pedons by harvest simulation type with weathering rates from Warba series. (**a**–**e**) WTHs decreased the total number of rotations for each nutrient as compared to SOH, with large variability in the systems shown by confidence intervals ($\alpha = 0.05$), and only the K pool was depleted. (**a**,**b**) N, P, and Mg showed minimal impacts through WTH and SOH. (**c**) K showed depletion after 3 WTHs and 4 SOHs. (**d**) Ca showed differences between SOH and WTH halfway through the model Harvest Recovery Years.

**Table 3.** Recovery time in years for weathering and deposition inputs to replace outputs of a single harvest and leaching by harvest type. The number of recovery years demonstrates that the inputs do not replace outputs at current rotation lengths, besides K and Mg in the non-texture-adjusted ceiling and average of the maximum weathering rates. * indicates the weathering rates and deposition do not exceed the outputs indicating that the harvest nutrient pool is not replaced.

| Soil Type | Weathering Type | Rate | SOH | | | WTH | | |
| --- | --- | --- | --- | --- | --- | --- | --- | --- |
| | | | K | Ca | Mg | K | Ca | Mg |
| Entic Haplorthods/Typic Udipsamments | Outwash Soils (Kolka et al.,1996) | Max. | 282 | 128 | 54 | 379 | 163 | 71 |
| | | Avg. | * | 186 | 90 | * | 238 | 119 |
| | | Min. | * | 448 | 276 | * | 573 | 367 |
| | Texturally Adjusted to Observed | Max. | * | 291 | 175 | * | 372 | 233 |
| | | Avg. | * | 403 | 335 | * | 516 | 444 |
| | | Min. | * | 696 | * | * | 890 | * |
| Medium to Nutrient-rich | Warba (Kolka et al., 1996) | Min. | * | 107 | 77 | * | 137 | 102 |

## 4. Discussion

### 4.1. Harvest Removals

The annual inputs of K, Ca, and Mg to the aspen ecosystems over 45 years did not replace the nutrients lost to harvest. On the other hand, pools of nutrients associated with organic matter (mainly N and P) were minimally impacted by WTH in this study, which is consistent with extensive modeling efforts conducted in the same ecosystem [51]. This current study highlights the importance of base cations present in biomass removal. Even under maximum rates of base cation release from mineral weathering, WTH shortened the total number of rotations until depletion for the K and Ca (4 WTHs; 5 SOHs) and Mg pools (6 WTHs; 7 SOHs). The larger context of nutrient-poor soils represented by the NCSS restricted pedons under maximum weathering rates showed depletion for K (4 WTHs; 5 SOHs) and Ca (5 WTHs; 7 SOHs). Ca and Mg pools have long been reported as a potential concern using WTH in aspen stands [4,5] and especially on coarse-textured soils [7]. Calcium translocation from the deeper horizons has been proposed as an unmeasured input [52]. Although there are differences in bedrock types in these studies, if the relatively Ca-rich values from prior work [52] were incorporated into the harvest models in this study, the recovery years would be 47–135 for SOH and 60–173 for WTH, on soils 10 m from bedrock. This would indicate that only the highest values from the shallowest bedrock would approach satisfying removals on an average rotation length. In the absence of increased nutrient inputs to the soils, the recovery ages under the maximum weathering rates exceed the rotation lengths of aspen.

The harvest simulations call into question the practice of continual SOHs and WTHs on the sandy outwash soils in the absence of increasing the input terms of K, Ca, Mg, and also on the Alfic Haplorthods without increased inputs of K. Our approach in this assessment was conservative in two ways: (i) outputs to the system through harvest removals and leaching were smaller than previously reported (cf., [53]; Table A3), and (ii) inputs through weathering and deposition were parametrized across the range of possible values, and an additional input term of nutrient translocation was provided. There are some reasons for inconsistencies across studies evaluating base cation balances in response to harvesting. Intensively managed forests can be further along in the harvest rotation models or have lower starting soil pool values than the average field observed soil values (for example, in this study, 989 K, 3217 Ca, and 336 Mg kg ha$^{-1}$). For instance, if the 30 cm pools from the observed WTH and SOH are extrapolated to 150 cm, the average exchangeable soil nutrients would be SOH (3650 K, 2100 Ca, and 1850 Mg kg ha$^{-1}$) and WTH (3650 K, 550 Ca, and 1650 Mg kg ha$^{-1}$) ([10]; Table A3). Extrapolating the top 30 cm values to 150 cm will overestimate the soil macronutrient values since the 30 cm values are comprised of the A-B horizons, which have larger macronutrient pools than the lower soil horizons. The average Ca value reported after WTH has important consequences because the next harvest

removal, regardless of the type, has the potential to deplete Ca. These results indicate that some outwash soils may be more at risk for negative harvest impacts depending on the management history and weathering status of the aeolian soil mass.

### 4.2. Long-Term Nutrient Budgets Comparison to Short-Term Harvest Intensity Experiments

The goal of this study was to model long-term patterns in the ecosystem and may not depict short-term phenomena well due to the absence of decomposition models. Although the outputs of the ecosystems through both harvest and leachate are smaller in this study than previously reported, the harvest time series show major losses to the outwash soils through SOH and WTH and shorter recovery years than rotation lengths for both soil types. In nutrient budget studies, WTH has been identified as having the potential to remove two to three times the nutrients as SOH [6], with larger impacts on coarse-textured soils [7], and Ca and Mg identified as at risk from WTH on more productive soils [4,5]. The short-term ($\leq$1 rotation) observations of the soil pools post-harvest found non-significant or site-specific impacts between WTH and SOH [10,11]. However, the models did show patterns of larger WTH removals than SOH that decreased available nutrients, but significant differences may not be observed until after multiple sequential WTHs (Figures 3–5). The long-term effects of harvest intensity have been suspected since mature aspen has accumulated the most nutrients [17,54], and our models confirmed that pattern.

Complications of detecting differences in the exchangeable soil pools between SOH and WTH in the short term include seasonal water table fluctuations, nutrient translocation to foliage [10], and root decomposition [11]. The harvest confidence intervals show that significant differences may not be detected until multiple sequential harvests have occurred, and the variation within the ecosystem harvest intervals is large due mainly to the soil values. The value of the harvests, SOH 339 N, 36 P, 204 K, 779 Ca, 63 Mg (kg ha$^{-1}$) and average WTH 428 N, 50 P, 272 K, 999 Ca, 84 Mg (kg ha$^{-1}$), and the additional volume removed from WTH 89 N, 14 P, 68 K, 219 Ca, 21 Mg, tend to be smaller or comparable to the standard error of the soils: NRCS Entic Haplorthods and Typic Udipsamments 506 N, 175 P, 123 K, 339 Ca 39 Mg (kg ha$^{-1}$), NCSS Entic Haplorthods and Typic Udipsamments 996 N, 260 P, 65 K, 399 Ca, 67 Mg (kg ha$^{-1}$), NCSS Alfic Haplorthods 433 N, 109 P, 90 K, 1131 Ca, 519 Mg (kg ha$^{-1}$). Barring having any pre-harvest measurements of soil nutrient pools, differences in harvest type appear difficult to detect in soils after a single harvest, although differences in removal have been observed in harvest residues [19]. These data indicate that the detection of soil nutrient depletion in response to harvest systems may take several rotations, depending on soil type or harvest intensity.

### 4.3. Modeled Ecosystem Values

The harvest removals reported herein were consistent with previously reported aspen harvest values ([5]; Table A3) and slightly above the reported harvest at 30-year estimation ([4]; Table A3). In comparison to the most similar outwash soils, both the sampled and NCSS outwash pedons fall between extrapolated Entic Haplorthods [10] and the Typic Udipsamments [53], except for the Mg, which fell below both, but remained near a Typic Haplorthods ([55]; Table A3). The differences in soil nutrient values within the Entic Haplorthods and Typic Udisamment studies are likely due to the textural variation within the outwash soils and the depletion status of the aeolian mass. Given the differences in soil type and potential overextrapolation of the literature soil values due to differences in sampling depths, the soil values reported in this study fall within historical values (Table A3) for the glacial till and outwash soils. Additionally, the leaching values from the water budget approach are more conservative than have been previously reported ([53]; Table A3). The leaching rates measured at 60 cm [53] and 40 and 130 cm [5] may have over-estimated leaching losses in prior studies because these depths are still within the zone of active root uptake (150 cm) [24].

### 4.4. Desired Future Condition

It is important to remember many of the soils were not agriculturally productive and considered the "lands nobody wanted" [1], and exist now as publicly owned forest lands. The low fertility of the sandy soils in combination with the modeled outputs exceeding inputs at the current rotation length indicated that future yields could be diminished if the soil becomes depleted. The impacts from the increased removal of nutrients should manifest in decreased productivity when a single nutrient reaches the limiting level [56]. Declines in productivity have been attributed to WTH and removal of the forest floor after 20 years in coarse-textured soil [11], and yet differences in forest productivity were not observed after one WTH rotation [18]. These results indicate, in some locations, nutrients have not become limiting to the aspen, which is adapted to nutrient-poor soils. If such a point of depletion were reached, for example, after 3–6 rotations of intensive harvesting with no increases to the input terms, the sandy outwash ecosystem could be expected to convert to pine or oak savannahs and function more similarly to a barrens ecosystem. If the cations became limiting, the yields could be expected to decrease and reduce the harvest output terms sequentially. The modeling indicates that harvest removals outpace the inputs to the system. Since the recovery years are longer than the biological and economic rotation of aspen, a clear answer is to increase the inputs if harvest removals keep pace with current levels [6]. Together these analyses indicate that current harvest levels are not sustainable if the desired future condition of these forests is aspen. These findings can be used in evaluating the desired future condition of intensively harvested aspen forests (in transitioning to oak or pine barrens ecosystems).

### 5. Conclusions

Under all weathering scenarios, the recovery years from a SOH or WTH exceeded the average rotation length (45 years) of the aspen. Simulations showed that increased harvest intensity from SOH to WTH of aspen on nutrient-poor soils shortened the available nutrient pools on average by one full rotation length for Ca, Mg, and K, and K for the slightly richer Alfic Haplorthods. N and P were less impacted for both soils since the percent removed in each harvest was generally small in comparison to the residual tissues and soil pools. The harvest simulation models called into question the sustainability of the SOH practice on nutrient-poor soils given the impacts to Ca, Mg, and K, and for K in the Alfic Haplorthods. The seasonal differences between WTH removals were small in comparison to the difference between WTH and SOH. The large portion of base cations located within the vegetative pools implies that inputs by weathering and deposition could not keep pace with harvest removals within the sandy, nutrient-poor soils in northern Wisconsin. Although there was much uncertainty due to variation in aeolian mass, weathering depletion status, and harvest removals, our models indicate that the NRCS sampled soils would be depleted in K, Ca, and Mg in four or five WTHs and five or six SOHs. The NCSS outwash soils were depleted within three (K), five (Ca), and ten (Mg) WTHs and four (K) to six (Ca) SOHs, and the NCSS Alfic Haplorthods were depleted of K within three WTHs and four SOHs. The long-term models bridged the gap between short and long-term studies and showed the harvest intensity effects might not be apparent until multiple harvests have occurred.

**Author Contributions:** Conceptualization, R.P.R., E.S.K., D.R.B. and R.K.K.; methodology, R.P.R., E.S.K., D.R.B. and R.K.K.; software, R.P.R.; validation, R.P.R., E.S.K. and D.R.B.; formal analysis, R.P.R., E.S.K. and D.R.B.; investigation, R.P.R., E.S.K., D.R.B. and R.K.K.; resources, E.S.K. and D.R.B.; data curation, R.P.R.; writing—original draft preparation, R.P.R., E.S.K. and D.R.B.; writing—review and editing, R.P.R., E.S.K., D.R.B. and R.K.K.; visualization, R.P.R.; supervision, E.S.K. and D.R.B.; project administration, E.S.K. and D.R.B.; funding acquisition, E.S.K. and D.R.B. All authors have read and agreed to the published version of the manuscript.

**Funding:** This research was funded by the Wisconsin Department of Natural Resources, in-kind support from the U.S. Forest Service, Northern Research Station, and the Ecosystem Science Center.

**Institutional Review Board Statement:** Not applicable.

**Informed Consent Statement:** Not applicable.

**Data Availability Statement:** The source data can be found in the public repository https://zenodo.org/record/4831614#.YLBbrPlKiUk and DOI: 10.5281/zenodo.4831614 accessed on 28 May 2021.

**Acknowledgments:** The Wisconsin Department of Natural Resources provided personnel (Teresa Pearson and Laura Reuling), funding, equipment, and field sampling, in conjunction with the U.S. Forest Service providing field technicians. In-kind support was received from the U.S. Forest Service, Northern Research Station. The Natural Resource Conservation Society soil scientists Ryan Bevernitz, Alexander Gajdosik, and Scott Eversoll provided soil expertise, field training, and sampling. Thanks to Stan Vitton for the early revision comments. The field and lab technicians that made the project possible: Calvin Norman, Eddie Hodges, Chelsea Bach, Jack Zwart, Sam Claire, Deanna Siel, Anya Leach, Joel Taylor, Jake DeVries, Jesse Barta, and Kaitlyn Dodge.

**Conflicts of Interest:** The authors declare no conflict of interest.

## Appendix A

**Table A1.** Soil texture within the top 150 cm comparing observed soil pedons to outwash soils used to calculate weathering rates. The named soils which show larger clay and silt percentages [24].

| Soil Type | Pedon ID | Clay (%) | Silt (%) | Sand (%) |
|---|---|---|---|---|
| Restricted | S2016WI037002 | 1.3 | 7.0 | 91.7 |
| | S2016WI037004 | 0.8 | 5.0 | 94.2 |
| | S2016WI051002 | 1.8 | 5.1 | 93.1 |
| | S2016WI075001 | 1.9 | 7.2 | 90.9 |
| | S2016WI075002 | 1.8 | 6.5 | 91.7 |
| | S2016WI085004 | 1.3 | 4.8 | 90.5 |
| | S2016WI085005 | 2.7 | 8.5 | 88.8 |
| | S2016WI085006 | 1.0 | 8.2 | 90.8 |
| | S2016WI085007 | 1.5 | 6.1 | 92.3 |
| | S2016WI113002 | 1.5 | 2.8 | 95.7 |
| | S2016WI129001 | 2.5 | 4.3 | 93.2 |
| | NRCS pedons | $1.66 \pm 0.18$ | $5.95 \pm 0.52$ | $92.09 \pm 0.57$ |
| | NCSS pedons | $1.81 \pm 0.13$ | $5.87 \pm 0.74$ | $92.30 \pm 0.80$ |
| | Cloquet | 7.5 | 16.7 | 75.8 |
| | Omega | 5.7 | 8.9 | 85.3 |
| | Solon Springs | 7.2 | 3.3 | 89.5 |
| Non-restricted | NCSS pedons | $5.97 \pm 0.43$ | $28.25 \pm 1.66$ | $64.36 \pm 2.46$ |
| | Warba | 21.3 | 22.2 | 56.5 |

**Table A2.** Augmented Dickey–Fuller unit root test *p*-values indicating most processes are white-noise ($\alpha = 0.05$) after differencing and log transformation. All K values fail the test, however, when excluding the 2017 values the processes appear to be detrended or white-noise.

| Region | N | K | Ca | Mg |
|---|---|---|---|---|
| Northeast | 0.05 | 0.48 | 0.01 | 0.01 |
| Northcentral | 0.02 | 0.41 | 0.02 | 0.01 |
| Northwest | 0.05 | 0.06 | 0.07 | 0.03 |

**Table A3.** Literature values of ecosystem nutrients in the Great Lakes region extrapolated from the sampling depth to 150 cm.

| Reference | Site Information | Depth (cm) | Type | N | P | K | Ca | Mg |
|---|---|---|---|---|---|---|---|---|
| | | | | kg ha$^{-1}$ | | | | |
| Alban et al., 1978 | Warba (Haplic Glossudalfs); loamy calcareous glacial till on moraines | 36 | - | 8575 | 358 | 1225 | 11,013 | 1108 |
| Boyle et al., 1973 | Iron River (Alfic Fragiorthods), Monico (Typic Endoaquods); silty or loamy deposits, overlying sandy loam till | 6 | - | 673 | 2858 | 3222 | 16,420 | |
| Perala and Alban, 1982 | Warba (Haplic Glossudalfs), loamy calcareous glacial till on moraines | 61 | - | 7766 | 268 | 2855 | 24,084 | 4281 |
| Perala and Alban, 1982 | Unnamed loamy fine sand | 61 | - | 6071 | 344 | 814 | 7247 | 969 |
| Pastor and Bockheim, 1984 | Pence (Typic Haplorthods), thin loamy alluvium mantle overlying stratified sand | 30 | - | 16,100 | 508 | 705 | 4200 | 590 |
| Premer et al., 2019 | Rubicon (Entic Haplorthods), sandy glaciofluvial deposits | 30 | SOH | 14,050 | - | 3650 | 2100 | 1850 |
| | | | WTH | 15,700 | - | 3650 | 550 | 1650 |
| | Onaway (Inceptic Hapludalfs), loamy deposits on moraines and drumlins | | SOH | 18,050 | - | 3000 | 5450 | 2550 |
| | | | WTH | 17,450 | - | 4200 | 5500 | 5750 |
| Silkworth and Grigal, 1982 | Newfound (Typic Fragiudepts), gravelly noncalcareous sandy loam glacial till | 130 | - | - | - | 741 | 10,478 | 2153 |
| Wilhelm et al., 2013 | Grayling/Menahga (Typic Udipsamments), Omega (Typic Haplorthods); sandy glaciofluvial deposits | 60 | | 190 | 1278 | 875 | 6200 | 1603 |
| *Aspen WTH Removal* | | | | | | | | |
| Silkworth and Grigal, 1982 | | 130 | - | 454 | 43.1 | 354.6 | 1034 | 94.5 |
| Boyle et al., 1973 | (Harvest at 30 years) | 6 | - | 172 | 24 | 116 | 382 | - |
| *Leaching* | | | | kg ha$^{-1}$ 45 year$^{-1}$ | | | | |
| Silkworth and Grigal, 1982 | | 130 | - | 18 | 25 | 131 | 1424 | 604 |
| Wilhelm et al., 2013 | | 6 | - | 138 | 85 | 170 | 344 | 353 |

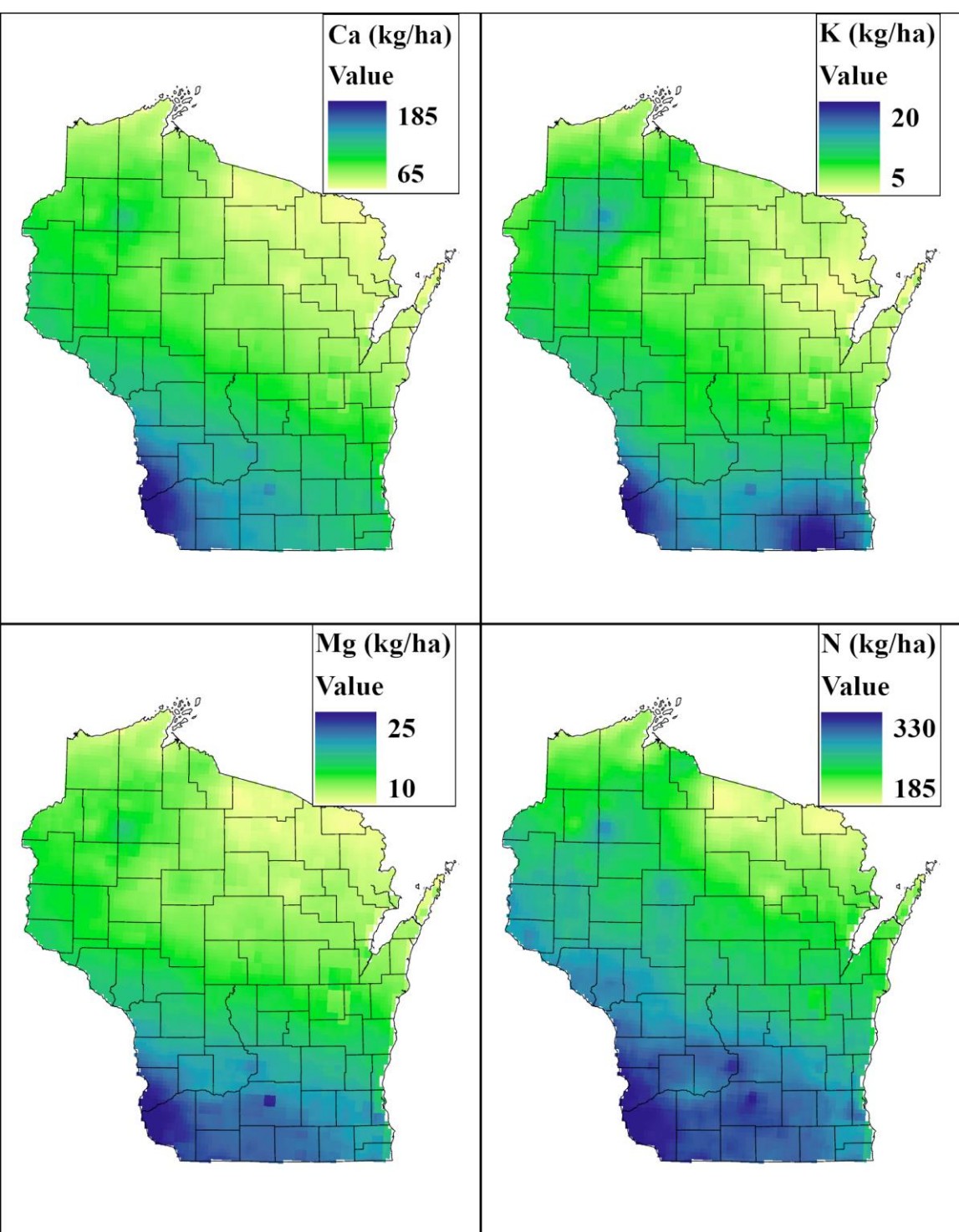

**Figure A1.** Total atmospheric deposition (wet + dry) throughout one 45-year rotation (adapted from [25]) of nutrients across the state of Wisconsin. In general, the south and southwestern areas are receiving two to three times the amount in the northeastern area.

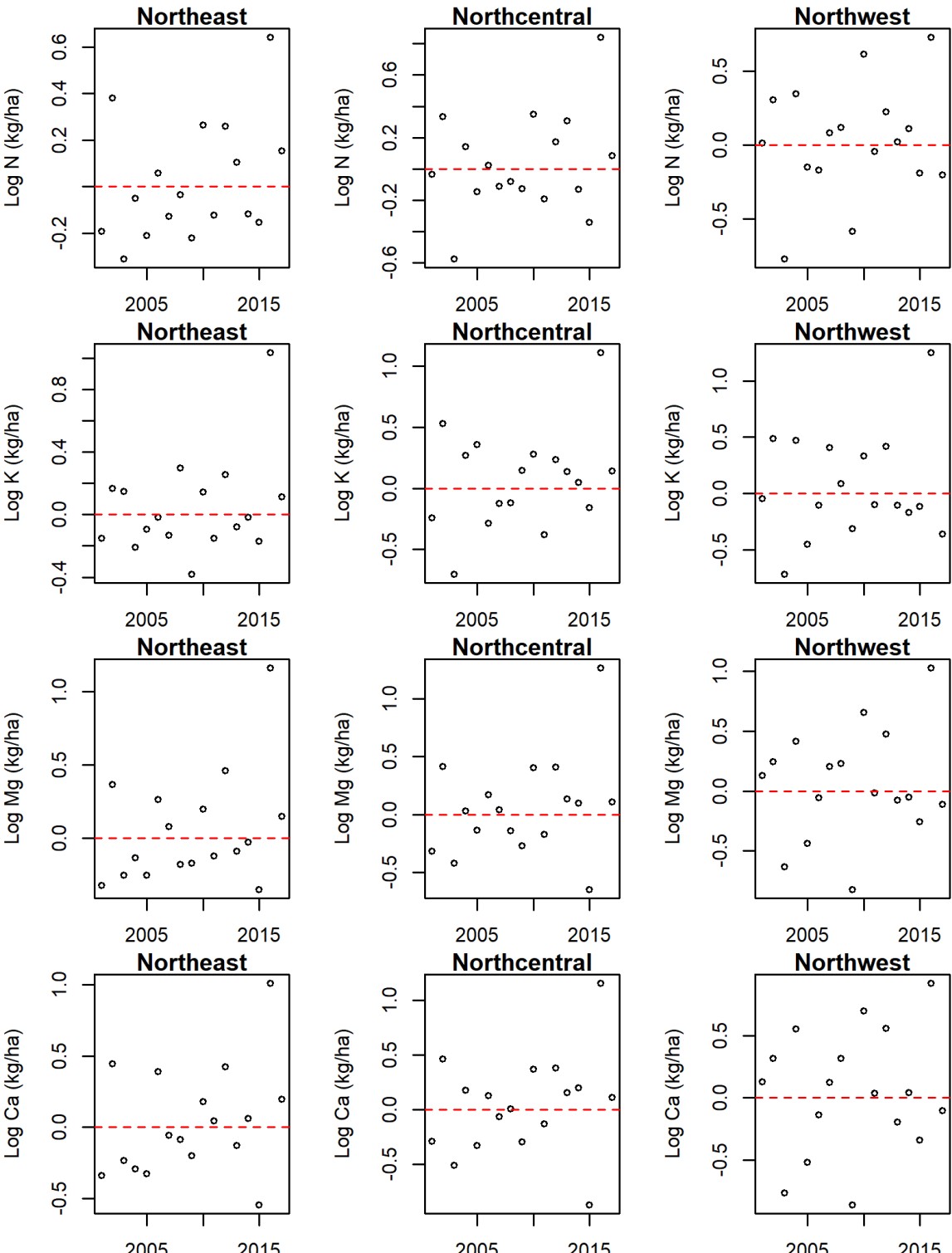

**Figure A2.** Time series of total atmospheric deposition time series for regional harvest model inputs after differencing and log transformation. When excluding the large spike in the 2017 deposition, seen as the second from right value due to differencing, the processes appear to generally be detrended or white-noise.

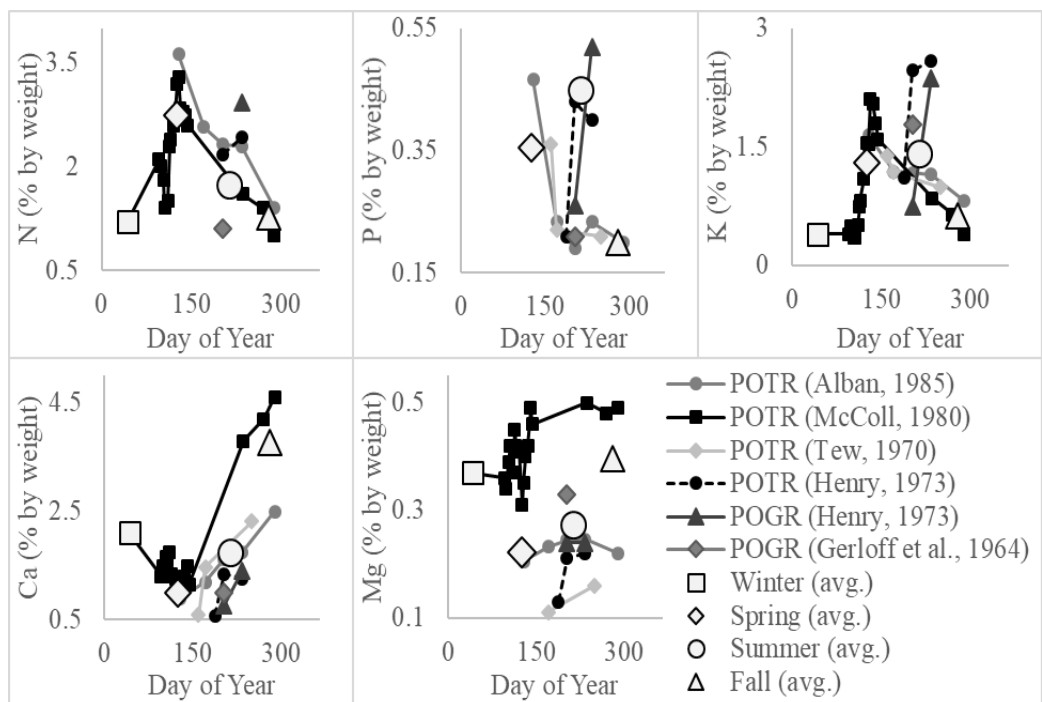

**Figure A3.** Seasonal foliar macro-nutrient concentrations by day of the year observed in aspen, as well as the average of the seasonal averages across studies (Derived from [13,46–49]) N, P, and K show increases during the growing season while decreasing during the dormant season, with Ca and Mg showing the opposite pattern. Trembling aspen (*Populus tremuloides*) is represented by POTR, and bigtooth aspen (*Populus grandidentata*) is represented by POGR.

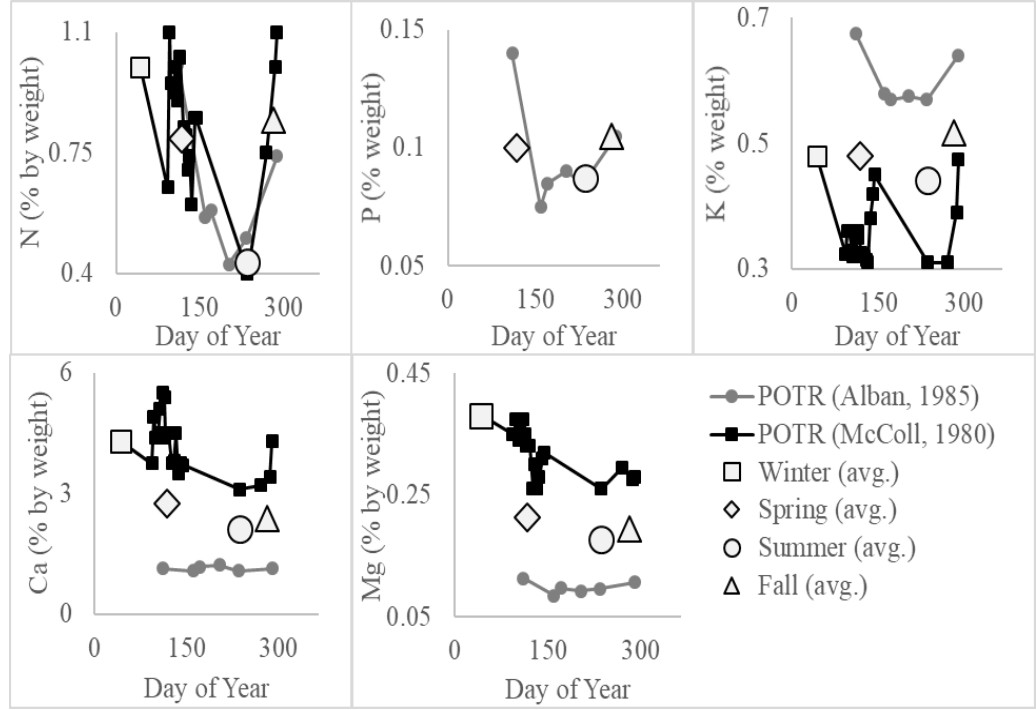

**Figure A4.** Seasonal twig macro-nutrient concentrations by day of the year observed in aspen, as well as the average of the seasonal averages across studies (Derived from [13,46–49]). The macro-nutrients all show increases during the dormant season while decreasing during the growing season. Trembling aspen (*Populus tremuloides*) is represented by POTR, and bigtooth aspen (*Populus grandidentata*) is represented by POGR.

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
