# Peer review of "Long-Term Ecosystem Nutrient Pool Status for Aspen Forest Harvest Simulations on Glacial Till and Sandy Outwash Soils"

_forests, doi:10.3390/f12111556_

Round 1

Reviewer 1 Report

The present article stands out in almost all areas used for review:

The topic is of interest, especially in today’s society and when looking towards future practices and improvements. 

Furthermore, the article bridges two important directions used until now in previous studies and in forest management policies. What stand out are the relevant data presented in the article as well as the impressive research sustained by a good methodology and relevant articles. 

The conclusions come as a concise review of the obtained data and responds to the previously mentioned hypotheses. The only recommendation that I see is that they should also include a hint about how the research can be applied in other situations/location or extended in the future. 

The article is very well written, with attention to both phrases and the style used.

Author Response

Reviewer 1

The present article stands out in almost all areas used for review:

The topic is of interest, especially in today’s society and when looking towards future practices and improvements. 

Furthermore, the article bridges two important directions used until now in previous studies and in forest management policies. What stand out are the relevant data presented in the article as well as the impressive research sustained by a good methodology and relevant articles. 

The conclusions come as a concise review of the obtained data and responds to the previously mentioned hypotheses. The only recommendation that I see is that they should also include a hint about how the research can be applied in other situations/location or extended in the future. 

The article is very well written, with attention to both phrases and the style used.

Response: We thank the reviewer for the recommendation about extending this research to future consideration (Line 507-510). Together these analyses indicate that current harvest levels are not sustainable, if the desired future condition of these forests is aspen. These findings can be used in evaluating the desired future condition of intensively harvested aspen forests (in transitioning to oak or pine barrens ecosystems). 

Reviewer 2 Report

Dear Authors,

Your contribution is of high scientific level, good structured, and with highly valuable output.

Some minimal additional explanations and broader interpretation would be useful.

Line 25

What was the number of sequential harvests you studied? The corresponding information you have presented in the text, but it should be useful to mention your important long-term results in the Abstract too.

Line 387

When the weathering rates were adjusted to textures observed in the field, the recovery years increased greatly and further demonstrated the non-sustainable removal rates for the outwash soils (Table 3).

It seems like some controlled force is capable to adjust the process you describe. But this is a natural process we can’t influence. It should be better to exclude the slight inconvenience in the text.

Line 500

…function more similarly to a barrens ecosystem

Yes. This is true. And not only for the ecosystem you studied. I support your concern.

Lines 503-505

Since the recovery years are longer than the biological and economic rotation of aspen, a clear answer is to increase the inputs if harvest removals keep pace with current levels.

Yes. You are right. The standard economical approach contradicts the biosphere. The origin of contradiction is the fact that the business is seeking for the short-term alternatives to reduce the expenses and increase the profit. This contradicts to the biosphere, which has no alternative. Your research is a valuable contribution to the last point of view. The simplest action: a return of matter to the biogeochemical cycle became a problem now. A possible way out is our propositions in the field of Biogeosystem Technique. I should be glad if you would cast a glance at https://doi.org/10.1016/j.envres.2020. I can send you a full-text article on your request.   

Lines 524-526

The long-term models bridged the gap between short and long-term studies and showed the harvest intensity effects may not be apparent until multiple harvests have occurred.

Yes. Then your next steps are desirable.

First. I propose you to mention the universal danger of the biosphere depletion you showed in example of the studied objects.

Second. It should be great if you propose in your paper a change of the current short-term business land-use paradigm, which is dangerous for the world development strategy.  

Author Response

Reviewer 2

Dear Authors,

Your contribution is of high scientific level, good structured, and with highly valuable output.

Some minimal additional explanations and broader interpretation would be useful.

Line 25

What was the number of sequential harvests you studied? The corresponding information you have presented in the text, but it should be useful to mention your important long-term results in the Abstract too.

Response: We thank the reviewer for this recommendation and have included it in the abstract (Line22). To assess the long-term sustainability, a nutrient budget was constructed from field measurements, the National Cooperative Soil Survey (NCSS) database, and literature values for stem-only harvesting (SOH) and WTH at a 45-year rotation length and 11 rotations were simulated.  

Line 387

When the weathering rates were adjusted to textures observed in the field, the recovery years increased greatly and further demonstrated the non-sustainable removal rates for the outwash soils (Table 3).

It seems like some controlled force is capable to adjust the process you describe. But this is a natural process we can’t influence. It should be better to exclude the slight inconvenience in the text.

Response: We thank the reviewer for bringing this to our attention. Since the weathering rate is related to the aeolian mass in the outwash soils and treated a model parameter in the study, we believe it is important to include the values to describe the full potential variation that was observed throughout the landscape to best depict potential conditions within the soil map units.

Line 500

…function more similarly to a barrens ecosystem

Yes. This is true. And not only for the ecosystem you studied. I support your concern.

Response: We thank the reviewer for highlighting this phrase. We are trying to predict what could happen with the ecosystem and although we share the concern for long-term soil nutrient status we also note that pine barren’s ecosystems are currently a management goal for many ecosystems in the area being studied due to their importance as wildlife habitat. The soils of interest in the study also have poor nutrient retention and commonly have barrens in addition to forests. Please see additional text in response to Reviewer 1 at line 507-510 which identifies potential future work that is outside of the scope of this study. 

Together these analyses indicate that current harvest levels are not sustainable, if the desired future condition of these forests is aspen. These findings can be used in evaluating the desired future condition of intensively harvested aspen forests (in transitioning to oak or pine barrens ecosystems).  

Lines 503-505

Since the recovery years are longer than the biological and economic rotation of aspen, a clear answer is to increase the inputs if harvest removals keep pace with current levels.

Yes. You are right. The standard economical approach contradicts the biosphere. The origin of contradiction is the fact that the business is seeking for the short-term alternatives to reduce the expenses and increase the profit. This contradicts to the biosphere, which has no alternative. Your research is a valuable contribution to the last point of view. The simplest action: a return of matter to the biogeochemical cycle became a problem now. A possible way out is our propositions in the field of Biogeosystem Technique. I should be glad if you would cast a glance at https://doi.org/10.1016/j.envres.2020. I can send you a full-text article on your request.   

Response: We thank the reviewer for this insight into the different time frames of the interacting systems. We believe it is best to not get into the specifics of management recommendations but instead document the nutrient status. This approach allows land managers the flexibility of deciding the right tactic for the local conditions. For example, a commonly used amendment after WTH in this type of soil is wood ash, but we prefer to avoid making any specific management recommendations to avoid limiting potential solutions. Increasing nutrient inputs to the system could be used to attempt to sustain yields at current rates, however, managers may decide it would be useful to convert the land to barrens. Currently the maintenance of barrens requires burning or non-commercial aspen management in some areas. Due to the complexity of land management in local scenarios and the potential solutions available we prefer to remain agnostic in terms of recommendations.    

Lines 524-526

The long-term models bridged the gap between short and long-term studies and showed the harvest intensity effects may not be apparent until multiple harvests have occurred.

Yes. Then your next steps are desirable.

First. I propose you to mention the universal danger of the biosphere depletion you showed in example of the studied objects.

Second. It should be great if you propose in your paper a change of the current short-term business land-use paradigm, which is dangerous for the world development strategy.  

Response: We thank the reviewer for the recommendations but believe the topics are outside of our expertise and that our manuscript could be used by land managers, economists, and policy makers to solve the issues mentioned.